# Prevalence, serotype and antibiotic susceptibility of Group B *Streptococcus* isolated from pregnant women in Jakarta, Indonesia

Dodi Safari[1], Septiani Madonna Gultom[2], Wisnu Tafroji[1], Athiya Azzahidah[3], Frida Soesanti📷[2], Miftahuddin Majid Khoeri[1], Ari Prayitno[2], Fabiana C. Pimenta[4], Maria da Gloria Carvalho[4], Cuno S. P. M. Uiterwaal[5], Nina Dwi Putri📷[2]*

1 Eijkman Institute for Molecular Biology, Jakarta, Indonesia, 2 Faculty of Medicine, Department of Child Health, Universitas Indonesia/Cipto Mangunkusumo General Hospital, Jakarta, Indonesia, 3 Aix-Marseille University, Marseille, France, 4 Division of Bacterial Diseases, National Center for Immunization and Respiratory Diseases, Centers for Disease Control and Prevention, Atlanta, GA, United States of America, 5 Julius Center for Health Sciences and Primary Care, Julius Global Health, University Medical Center, Utrecht, the Netherlands

* ninadwip@gmail.com

**Data Availability Statement:** All relevant data are within the paper.

**Funding:** This study was partly funded by Partnerships for Enhanced Engagement in

## Abstract

Group B *Streptococcus* (GBS) is a bacterial pathogen which is a leading cause of neonatal infection. Currently, there are limited GBS data available from the Indonesian population. In this study, GBS colonization, serotype distribution and antimicrobial susceptibility profile of isolates were investigated among pregnant women in Jakarta, Indonesia. Demographics data, clinical characteristics and vaginal swabs were collected from 177 pregnant women (mean aged: 28.7 years old) at 29–40 weeks of gestation. Bacterial culture identification tests and latex agglutination were performed for GBS. Serotyping was done by conventional multiplex PCR and antibiotic susceptibility testing by broth microdilution. GBS colonization was found in 53 (30%) pregnant women. Serotype II was the most common serotype (30%) followed by serotype III (23%), Ia and IV (13% each), VI (8%), Ib and V (6% each), and one non-typeable strain. All isolates were susceptible to vancomycin, penicillin, ampicillin, cefotaxime, daptomycin and linezolid. The majority of GBS were resistant to tetracycline (89%) followed by clindamycin (21%), erythromycin (19%), and levofloxacin (6%). The serotype III was more resistant to erythromycin, clindamycin, and levofloxacin and these isolates were more likely to be multidrug resistant (6 out of 10) compared to other serotypes. This report provides demographics of GBS colonization and isolate characterization in pregnant women in Indonesia. The results may facilitate preventive strategies to reduce neonatal GBS infection and improve its treatment.

## Introduction

*Streptococcus agalactiae* (Group B *Streptococcus*; GBS) is a Gram positive coccus pathogen that can colonize the gastrointestinal and vaginal epithelium of healthy women potentially causing

Research (PEER) program - the U.S. Agency for International Development (Grant Number: 161463), Ministry of Research and Technology, Republic of Indonesia/National Research and Innovation Agency, the U. S. Global Health Security funds, and Publikasi Terindeks Internasional (PUTI) Universitas Indonesia. The funder had no role in study design, data analysis, decision to publish and preparation of the manuscript. The findings and conclusions in this report are those of the authors and do not necessarily represent the official position of the Centers for Disease Control and Prevention.

**Competing interests:** The authors have declared that no competing interests exist.

ascending intrauterine infection or transmission during parturition and creating a risk of serious disease in the vulnerable newborn [1]. This pathogen is a leading contributor to adverse maternal and newborn outcomes, with at least 409000 (uncertainty range [UR], 144000–573000) maternal/fetal/infant cases and 147000 (UR, 47000–273000) stillbirths and infant deaths annually, especially in Africa, where 54% of estimated cases and 65% of all fetal/infant deaths occur [2]. In high-income countries, GBS is well-known as one of the leading causes of infant deaths, especially in the first week of life [3]. Meanwhile, GBS disease burden estimation in low and middle income countries is more difficult due to several factors, such as a portion of births occur outside of hospital settings; failure to seek care; poor access to care; and healthcare facilities might lack laboratory capacity or resources to diagnose GBS infection [4].

In general, the prevalence of maternal GBS colonization was 18% worldwide with regional variation (11–35%) and 98% of identified colonizing GBS isolates were serotype Ia, Ib, II, III and IV worldwide whereas serotype III was associated with invasive disease [5]. In rural Mozambique, GBS serotype III was the most common serotype isolated from infants <90 days with invasive bacterial disease [6]. Meanwhile, GBS serotype Ia was reported as the most common serotype (40%) isolated from pregnant women in Bangladesh. Moreover, the newborn showed positive GBS (38%) were reported identical with mother colonizing GBS showing that vertical transmission was occurred to body sites of newborns [7]. Currently, data on GBS colonization and invasive bacterial disease are limited for the Indonesian population. In 1992, Wibawan et al reported that type pattern Ia/c and type pattern III and V were the main type patterns among 103 cultured GBS strains isolated from human clinical specimens such as urine, infected skin, sputum, sperm, and bronchial washings in Jakarta, Indonesia [8]. Previously, it was reported that GBS colonization was identified in 10 out of 38 pregnant women (26%) with or without complication in 2002, in Jakarta, Indonesia [9]. Budayanti and Sanjaya reported that GBS colonization rate was 31.3% among 32 pregnant women with gestational age 35–37 weeks in Denpasar, Bali, Indonesia in 2007–2008 [10]. In Southeast Asian region, the burden of GBS disease appears to be low and more rigorous studies are needed to determine if this is due to under-ascertainment or to true differences in the incidence of GBS disease in this region [11]. In this study, we evaluated the colonization, serotype distribution and antimicrobial susceptibility profile of GBS isolates recovered from pregnant women in Jakarta, Indonesia, and analyzed the association of clinical, social and demographic aspects with GBS carriage. We expect that our findings may contribute to new preventive strategies targeting GBS causing invasive bacterial diseases.

## Materials and methods

### Specimen collection

This study enrolled 177 pregnant women at 29–40 weeks of gestation from primary health centers in Jakarta, Indonesia. This study is a part of Effect of Air Pollution in Early Life on Infant and Maternal Health Project (https://sites.nationalacademies.org/PGA/PEER/PEERhealth/PGA_161463). The aim of the project is to establish relationships between pollutant exposure in pregnancy, and maternal and neonatal health indicators. Healthy pregnant women were enrolled and together with offspring were followed for six months after birth to record pregnancy complications, maternal lung function, neonatal indicators of fetal health, infant function, and others. This study was approved by The Research Ethics Committee of Faculty of Medicine, Universitas Indonesia/Dr. Cipto Mangunkusumo General Hospital, Jakarta, Indonesia (No. 895/UN2.F1/ETIK/2015). Vaginal swabs were collected from consented pregnant women at 29–40 weeks of gestation by trained nurses. Swabs were placed into 1 mL STGG (skim milk, tryptone, glucose, glycerol) media and transferred on wet ice to the Bacteriology

Laboratory at the Eijkman Institute for Molecular Biology, Jakarta. The specimens were stored at -80˚C until further analysis. The detail protocols were available at protocol.io (dx.doi.org/ 10.17504/protocols.io.bt6qnrdw).

## Bacterial isolation and identification

Vaginal swab specimens in STGG media were processed for GBS isolation and identification following 2010 CDC guidelines with modifications [12]. The inoculated STGG specimen was thawed, vortexed, and 200 μL volume was transferred into 5.0 mL of LIM broth with addition of 1.0 mL of rabbit serum (Gibco cat N. 16120099) and incubated at 37˚C, 5% $CO_2$, for 5 hours. A 10 μL aliquot of cultured broth was inoculated and streaked onto chromogenic agar plate (ChromID Strepto B) for colony isolation, then incubated at 37˚C, 5% $CO_2$, for 24–48 hours. A single presumptive GBS colony from each variation of pink colonies were sub cultured on 5% sheep blood agar plate (BAP) for ß-hemolysis, Gram staining and CAMP test [12,13]. Definitive grouping was done by latex agglutination (Remel cat no. R62031), according to the manufacturer's instructions. All confirmed GBS isolates were stored in STGG media at -80˚C for further analysis.

## Bacterial DNA extraction

The DNA was obtained using the fast extraction methodology for *Streptococcus* isolates [14]. Briefly, the extraction was done from an overnight growth pure culture of GBS isolates. The colonies were then harvested and resuspended into 300 μL of 0,85% saline. The suspension was incubated at 70˚C for 15 minutes followed by centrifugation at 10000 rpm for 2 minutes. Supernatant was removed and pellet was resuspended into 50 μL of TE buffer. A 12μL of 2500U/ml mutanolysin (Sigma) and 8μL of 30mg/ml hyaluronidase (Sigma) were added into bacterial suspension, incubated at 37˚C for 30 minutes followed by inactivation at 100˚C for 10 minutes. Final step, the lysates were centrifuged at 10000 rpm for 4 minutes. The DNA was then stored at 4˚C until further analysis.

## Serotyping of GBS isolates

Serotyping of GBS isolates was performed by conventional multiplex PCR [15]. Concisely, the serotyping was performed in multiplex PCR by using various concentration of each primer described previously. Mastermix reaction was prepared in 25μL containing 1X Qiagen Multiplex PCR mastermix (Qiagen), 19 primers at final concentration 0.25μM for each primer except for forward primer of cpsI-Ia-6-7 and cpsI-7-9 that were prepared at final concentration 0.4μM. DNA template used in this study was 2.5μl. Nuclease free water was added until 25μL. The PCR was done at following conditions: initial denaturation at 95˚C for 15 minutes, followed by 35 PCR cycles; denaturation at 94˚C for 30 seconds, annealing at 54˚C for 1 minutes 30 seconds, and elongation at 72˚C for 1 minute. Post-elongation was done at 72˚C for 10 minutes and hold at 4˚C. PCR products were visualized in 2% agarose gel at 100V for 90 minutes.

## Antimicrobial susceptibility testing

Antimicrobial susceptibility testing by broth microdilution was performed following the recommendations of Clinical and Laboratory Standards Institute minimum inhibitory concentration (MIC) breakpoints [16]. The antimicrobial susceptibility testing was performed against 14 antibiotics including erythromycin, clindamycin, cefazolin, vancomysin, penicillin, tetracycline, ampicillin, levofloxacin, cefotaxime, ciprofloxacin, cefoxitin, daptomycin, linezolid, and ceftizoxime prepared in customized 96 well plate according to CLSI breakpoints (Thermo

Sensititre[TM] Cat no. GAS-GBS CMC5CDCS). The antibiotic plate also provided inducible clindamycin resistant detection represented by a single well containing erythromycin/clindamycin (1/0.5). Inducible clindamycin was determined by the growth of GBS in this well.

## Data analysis

Data analysis was performed using Stata Software. Bivariate analysis was performed using Chi-Square and Fisher's Exact to evaluate the association between risk factors to colonization of GBS in pregnant woman. Logistic regression was performed to compute Odds Ratio (OR) with 95% Confidence Interval (CI).

Power of the test was computed using Stata Software. Previous study found the prevalent of GBS colonization in pregnant woman in Indonesia was 31% ($P_0$ = 0.31) [10] and we estimated an increase of 10% ($\partial$ = 0.1) therefor $P_a$ was 41%. Using these parameters, we found the power of the test for a sample size of 179 with alpha 0.05 was 74.3%.

## Results

### The prevalence rate of GBS

In this study, the prevalence of GBS among pregnant women Jakarta, Indonesia was 30% (53 out of 177). The GBS was isolated from vaginal swab specimen only collected from pregnant women. The characteristics of the pregnant women are described in Table 1. The maternal age range was 17 and 43 years (mean aged: 28.7 years old) and over 62% (n = 109) were aged 17–29 years. A total of 4% (7/177) of pregnant women had hypertension, only 3% (5/177) had pre-eclampsia, and 66% were multigravida (117/177). During the study period, 11% (20/177) and 16% (29/177) of the subjects had premature rupture of the membrane and history of spontaneous abortion respectively (Table 1). We observed that 71% (126/177) of the pregnant women had gestational age more than 37 week (at term) and 44% (78/177) of the subject delivered vaginally. We found there were no significant association between parity, gravida, hypertension, preeclampsia, premature rupture of the membrane, history of spontaneous abortion, gestational age, delivery method and birth weight with GBS colonization (Table 1). Newborns whose mothers were colonized with GBS had a mean of birth weight lower (3047 gram) than newborns whose mothers was not colonized (3222 gram) with p value = 0.013 (Table 1; Fig 1). Mothers colonized with GBS had mean of age older (30) than mothers who were not colonize with GBS (28) with p value = 0.01. Mothers aged 30 years and older were more likely to be colonized with GBS than mothers who were younger with OR 2.2 (95%CI: 1.1–4.2) (Table 1).

### Serotype distribution and antimicrobial susceptibility profile

A total of 53 GBS isolates were recovered from 177 vaginal swab specimens. Serotype II (30%; 16/53) was the most frequent followed by serotype III (23%; 12/53), serotype Ia and IV (13%; 7/53 each), serotype VI (8%; 4/53), serotype Ib and V (6%; 3/53 each). One GBS isolate (2%) was non-typeable using multiplex PCR. All GBS isolates were susceptible to vancomycin, penicillin, ampicillin, cefotaxime, daptomycin, and linezolid. Most GBS isolates were resistant to tetracycline (89%) followed by clindamycin (21%), erythromycin (19%), and levofloxacin (6%). All isolates resistant to erythromycin were also resistant to clindamycin and 9 of them were positive for the well of inducible resistance. GBS serotype III was more resistant to erythromycin, clindamycin, and levofloxacin. We observed that 19% (10/53) of GBS isolates expressed a resistance to three or more antibiotic agents of different classes that were defined as multidrug resistant (MDR) strains. We identified 6 out of 10 MDR isolates were serotype III followed by serotype Ib (2/10), serotype Ia (1/10), and serotype IV (1/10).

**Table 1. Demography and association between risk factors and GBS colonization among pregnant women.**

| Variables/Categories | n | maternal GBS colonization | | p-value | OR (95% CI) |
|---|---|---|---|---|---|
| | | Positive, n (%) | Negative, n (%) | | |
| **Maternal age (years)** | | | | | |
| 17–29 | 109 | 26 (23.9) | 83 (76.1) | ref | ref |
| 30–43 | 62 | 25 (40.3) | 37 (59.7 | 0.03 | 2.2 (1.1–4.2) |
| **Parity** | | | | | |
| Nullipara | 59 | 18 (30.5) | 41 (69.5) | ref | ref |
| Primipara | 59 | 17 (28.8) | 42 (71.2) | 0.840 | 0.9 (0.4–2.0) |
| Multiparous | 51 | 15 (29.4) | 36 (70.4) | 0.900 | 10.9 (0.4–2.2) |
| **Gravida** | | | | | |
| Primigravida | 52 | 15 (28.9) | 37 (71.1) | Ref | Ref |
| Multigravida | 117 | 35 (29.9) | 82 (70.1) | 0.888 | 1.1 (0.5–2.2) |
| **Hypertension** | | | | | |
| No | 141 | 44 (31.2) | 97 (68.8) | 0.342 | NA |
| Yes | 7 | 1 (14.3) | 6 (85.7) | | |
| **Preeclampsia/eclampsia** | | | | | |
| No | 143 | 45 (31.5) | 98 (68.5) | 0.323 | NA |
| Yes | 5 | 0 (0) | 5 (100.0) | | |
| **Premature rupture of membrane** | | | | | |
| No | 128 | 40 (31.3) | 88 (68.7) | Ref | Ref |
| Yes | 20 | 5 (25.0) | 15 (75.0) | 0.573 | 0.7 (0.3–2.2) |
| **History of spontaneous abortion** | | | | | |
| No | 133 | 39 (29.3) | 94 (70.7) | 0.584 | ref |
| Yes | 29 | 10 (34.5) | 19 (65.5) | | 1.3 (0.5–3.0) |
| **Gestational age at birth (week)** | | | | | |
| Term | 126 | 37 (29.4) | 89 (70.6) | Ref | Ref |
| Preterm | 5 | 2 (40,0) | 3 (60.0) | 0.613 | 1.6 (0.3–10.0) |
| **Delivery method** | | | | | |
| Vaginal | 78 | 22 (28.2) | 56 (71.8) | Ref | Ref |
| Vacuum | 5 | 1 (20.0) | 4 (80.0) | omitted | omitted |
| Cesarean Section | 63 | 21 (33.3) | 42 (66.7) | 0.511 | 1.3 (0.6–2.6) |
| **Maternal age (Mean, years)** | | 30.3 | 28.0 | 0.01 | NA |
| **Birth Weight (Mean, gram)** | | 3047.5 | 3222.9 | 0.013 | NA |
| **Birth Length (Mean, cm)** | | 48.2 | 48.4 | 0.616 | NA |

*NA = Not available.

## Discussion

Our data showed that the GBS colonization rate in pregnant women was 30% in Jakarta, Indonesia. This rate was higher than the other Asian countries mean rate of 12.8% (country variation: 8%-20%) and also higher than the global rate of GBS colonization (18%-26.2%) [5,17–19]. The true colonization rate may be higher since only vaginal swab was collected from the women. Studies have shown that collecting from both the lower vagina and rectum (through the anal sphincter) increases the culture yield substantially compared with sampling the cervix or the vagina without also swabbing the rectum [20]. Meanwhile, our data are similar to previous study that reported the prevalence of GBS colonization in Denpasar, Indonesia was 31.3% in 2007–2008 [10]. Moreover, in 2002, 26% of pregnant women with and without complications as such hemorrhage antepartum (n = 6), molar pregnancy (n = 3), and abortion (n = 2)

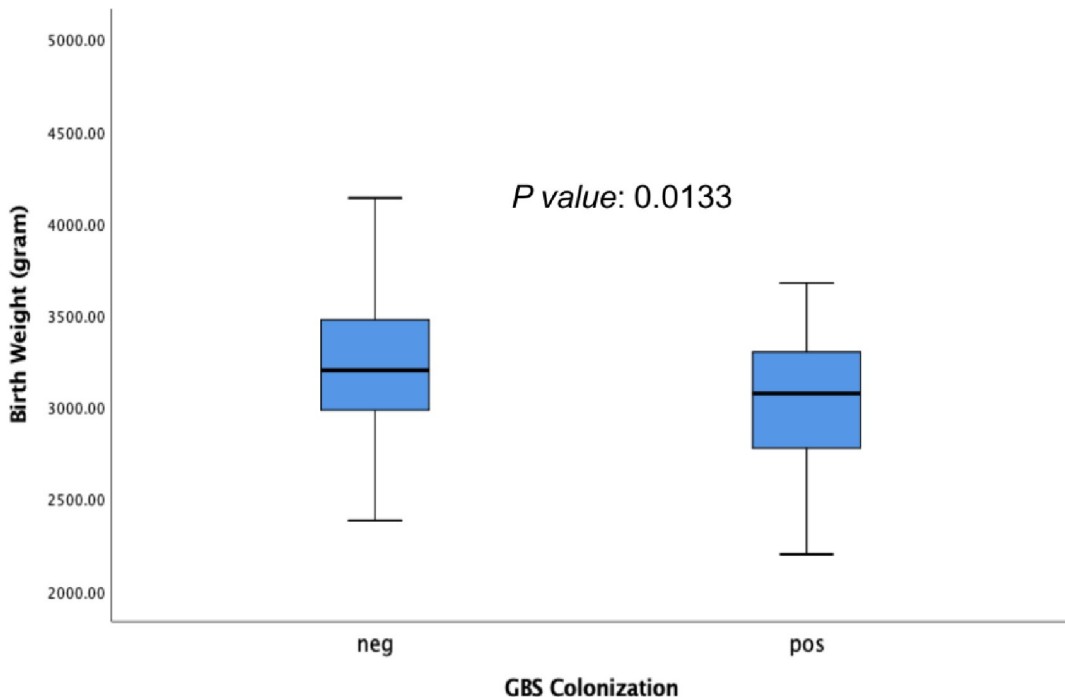

**Fig 1. Correlation between birth weight of newborns and mother with negative (neg) and positive (pos) for GBS colonization.**

(total participants: 38 women) from a similar area in Jakarta, were colonized with GBS [9]. In this study, we found that GBS colonized mothers were more likely to have newborns with lower birth weight (Fig 1). Maternal GBS colonization was previously reported to be associated with an increased risk of delivering a preterm, low-birth-weight infant [21].

Russell et al reported that GBS serotypes Ia, Ib, II, III, and V accounted for 98% of identified colonizing GBS isolates worldwide [5]. In Asia, serotype VI, VII, and VIII more frequently found [5]. Meanwhile, in our study, serotypes II, III, Ia, and IV accounted for 79% of all the GBS strains isolated from Indonesian pregnant women. Recently, Botelho et al reported that serotype II was the second most frequent serotype among 689 GBS isolates from pregnant women living in Rio de Janeiro, Brazil, over a period of eight years [19]. In African countries, the V, III, Ia, Ib, and II serotypes accounted for 91.8% of isolates [22]. Africa et al reported that in pregnant women from Western Cape, South Africa, the most common serotype were serotype V (67%), followed by serotype III (21%) [23]. In addition, among Gambian mothers, the predominant serotype was serotype V (55%), followed by II (16%), III (10%), Ia and Ib (each 8%) [24]. In another study in Toronto, Canada, the most frequently identified serotypes found among healthy pregnant women were III (25%), followed Ia (23%) and V (19%)[25]. In our study, we observed that the prevalence of serotype II was higher than other studies in Asia. In a systematic review, in mainland China, the five GBS serotypes (Ia, Ib, II, III, and V) accounted for the majority of the cases of GBS disease [21], but GBS serotype III was the dominant serotype observed colonizing pregnant women in Beijing [26]. Saha et al reported that serotype Ia was the most common serotype (40%) found in pregnant women in Mirzapur, Bangladesh, followed by serotype V (23%), II (14%), and III (12%) [7]. In Kuwait, serotype V was the most predominant serotypes among GBS isolates (39%) isolated from mothers, followed by III (21%), Ia (8%), and II (11%) [27].

All GBS isolates were still susceptible to penicillin, ampicillin, cefotaxime, vancomycin, daptomycin, and linezolid. However, we found that 89% of isolates were resistant to tetracycline. The susceptibility to penicillin found here is in line with previous studies on GBS colonization in pregnant women in different countries [12,25,27–30]. These data showed that penicillin might be the first choice of antibiotic for intrapartum prophylaxis and treatment of GBS infections [31]. In Indonesia, penicillin is still the most common antibiotic class consumed widely [32–34]. While there are occasional reports of penicillin resistance, resistance is so rare that penicillin can be safely used empirically [35]. At this moment, GBS screening was not part of routine prenatal care in Indonesia. Prophylactic antibiotics were commonly given prior caesarean section [36] and premature rupture of membrane. Recently, it was reported that pregnant women (14.5%) received antibiotics during the perinatal period with start dates ranged between 30 days to several hours prior to labour in Yogjakarta and Central Java Provinces, Indonesia with ceftriaxone and cefotaxime are common antibiotic use in that study [33].

Our study was limited to use vaginal swab specimens only instead of vaginal-rectal swab specimens to process GBS isolation. Moreover, we also used a stored specimen at -80˚C for approximately 5 months storage with no initial testing to identify viability of GBS. We declare these conditions as limitations of this study. However, during the isolation, we boosted the growth of GBS by adding rabbit serum as supplement during enrichment with the selective media LIM broth. We expected that the addition of rabbit serum followed by chromogenic agar plate culturing would boost GBS growth and help isolation. In conclusion, our study provides demographic and isolate characterization of GBS colonization in pregnant women. These results may facilitate potential preventive strategies to reduce neonatal GBS invasive infection and improve its treatment in Indonesia.

## Acknowledgments

We dedicated this work to Dr. Nikmah Salamia Idris (Former PI) who had passed away in 2020 –may she rest in peace. We are grateful to the pregnant women for participating in the study, all BRAVO team and the staff of the Department of Child Health, Dr. Cipto Mangunkusumo Hospital, Jakarta, Indonesia.

## Author Contributions

**Conceptualization:** Dodi Safari, Frida Soesanti, Ari Prayitno, Cuno S. P. M. Uiterwaal, Nina Dwi Putri.

**Data curation:** Septiani Madonna Gultom, Wisnu Tafroji, Athiya Azzahidah, Frida Soesanti, Miftahuddin Majid Khoeri.

**Formal analysis:** Dodi Safari, Wisnu Tafroji, Miftahuddin Majid Khoeri, Fabiana C. Pimenta, Maria da Gloria Carvalho.

**Investigation:** Nina Dwi Putri.

**Supervision:** Nina Dwi Putri.

**Writing – original draft:** Dodi Safari, Wisnu Tafroji, Athiya Azzahidah, Fabiana C. Pimenta, Maria da Gloria Carvalho.

**Writing – review & editing:** Dodi Safari, Wisnu Tafroji, Miftahuddin Majid Khoeri, Ari Prayitno, Fabiana C. Pimenta, Maria da Gloria Carvalho, Cuno S. P. M. Uiterwaal, Nina Dwi Putri.

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
