## [Decision Letter · Decision Letter 0]

3 Mar 2021

PONE-D-21-01468

Prevalence, serotype and antibiotic susceptibility of Group B Streptococcus isolated from pregnant women in Jakarta, Indonesia

PLOS ONE

Dear Dr. Putri,

Thank you for submitting your manuscript to PLOS ONE. After careful consideration, we feel that it has merit but does not fully meet PLOS ONE’s publication criteria as it currently stands. Therefore, we invite you to submit a revised version of the manuscript that addresses the points raised during the review process.

Methodology used needs more clarifications and discussions needs improvements. 

We look forward to receiving your revised manuscript.

Kind regards,

Iddya Karunasagar

Academic Editor

PLOS ONE

Journal Requirements:

2.We note that the grant information you provided in the ‘Funding Information’ and ‘Financial Disclosure’ sections do not match.

Additional Editor Comments:

The reviewers have raised a number of questions on methodology used and made suggestions for improvements in interpretation and discussion. Please address all reviewer comments point by point.

Reviewers' comments:

Reviewer's Responses to Questions

**Comments to the Author**

1. Is the manuscript technically sound, and do the data support the conclusions?

Reviewer #1: Partly

Reviewer #2: Partly

Reviewer #3: Yes

2. Has the statistical analysis been performed appropriately and rigorously? 

Reviewer #1: No

Reviewer #2: No

Reviewer #3: Yes

3. Have the authors made all data underlying the findings in their manuscript fully available?

Reviewer #1: Yes

Reviewer #2: Yes

Reviewer #3: Yes

4. Is the manuscript presented in an intelligible fashion and written in standard English?

Reviewer #1: Yes

Reviewer #2: Yes

Reviewer #3: Yes

5. Review Comments to the Author

Reviewer #1: This limited study is one of many studies describing GBS prevalence in pregnant population but one of the few from Indonesia. It is original in so far that describe the prevalence, serotypes and antimicrobial susceptibility in this population. The study is a part of a wider study 'Effect of Air Pollution in Early Life on Infant and Maternal Health Project'. It appears to be a single centre study. Statistics: No sample size calculation for estimating GBS prevalence undertaken. Thus it is not clear how the sample size was chosen. Only univariate statistics are presented when assessing association of GBS with maternal demographics, conditions and outcomes. No confidence intervals provided for rates. Microbiology: No rectal swabs taken. So likely to underestimate rates. Swabs taken and stored at -80C. However no assessment of viability of GBS. Authors have not stated how long swabs were stored before culture. It is not clear how resistance to clindamycin was determined. It is unusual for clindamycin resistance rates to be higher than erythromycin resistance rates. Was inducible resistance tested? The discussion is satisfactory but suggest rephrase the sentence in lines 152/3 to state the rate reported in this study may be an underestimation as only vaginal swabs were collected.

There is no section describing the limitations of the study. Finally the findings of this limited study may not be generalizable.

Reviewer #2: Overall – a straightforward study, with findings similar to previous data from Indonesia in 2010. While serotypes are interesting as potential vaccine targets, MLST or WGS would give more discriminating data, allowing more discriminating molecular epidemiology. The data need a statistical analysis.

Title – ok

Abstract – ok

Main text

1. Line 43 – I don’t consider GBS a ‘diplococcus’; surely just a coccus, or cocci in chains.

2. Line 49/50. I don’t understand this sentence. Perhaps you mean ‘... especially in Africa, where 54% of estimated cases and 65% of all 50 fetal/infant deaths occur.’

3. Line 56; this is a very long sentence, and therefore hard to read. I suggest you break it up into 2 sentences. ‘In general, the prevalence of maternal GBS colonization is 18% worldwide, with considerable regional variation (11-35%). Almost all (98%) colonizing GBS are serotype I-IV, whereas serotype III is associated with invasive disease [5].

4. Line 57; do you mean serotypes I and IV? .. or serotypes Ia, Ib, II, III and IV?

5. Line 60 -61. This sentence is not clear. Does this data on Ia refer to GBS pairs, isolated from both mother and child pairs? Does it refer to invasive or just colonising GBS? You must be very specific, otherwise the reader won’t understand your message. Are you telling us that Ia accounts for 40% of invasive neonatal disease? Simply pasting data from a reference without telling us why you are doing so, or interpreting the data for readers, is unhelpful.

6. Line 75 – change to ... ‘... findings may contribute to new preventive strategies targeting ...’

7. Line 90 – you could delete ‘The content and objectives of the study were explained to the participants and their consent obtained by signature on the appropriate consent forms.’.

8. Line 92; change ‘eligible’ to ‘consented’

9. Table 1 is confusing, as you have calculated % data in columns. Most readers are interested in the % of positive or negative colonisation in each group, meaning the % data should be across rows, not columns. I don’t think the ‘Frequency’ column needs % data at all.

10. Line 126; the text says that there were no differences in rates but I don’t see any statistical analysis (comparing carriage v non-carriage in each category).

11. Fig 1 does not show any statistical analysis.

12. Delete Table 2, as all the data is in the text – lines 138 – 146.

13. Line 149 - 152; a long sentence. It would be easier to read if you cut it up.

14. Line 152 - 153; I think you mean ‘The true colonisation rate ...’

15. Line 163 – The sentence includes serotype V, but doesn’t include IV, but earlier you said (Line 57), that I-IV accounted for 98% of GBS. Please harmonise.

16. Line 189; I think this is a little unfair, as a summary of the reference you cite; they actually say ‘While overall resistance is still relatively rare, a recent study highlighted the increased incidence of PR-GBS in Japan, increasing from 2.3% in 2005–2006 to 14.7% in 2012/13.’ A balanced view is along the lines of 'While there are occasional reports of penicillin resistance, resistance is so rare that penicillin can be safely used empirically'.

Reviewer #3: Manuscript #: PONE-D-21-01468

Title: Prevalence, serotype and antibiotic susceptibility of Group B Streptococcus isolated from pregnant women in Jakarta, Indonesia

Authors: Dodi Safari; Septiani Madonna Gultom; Wisnu Tafroji; Athiya Azzahidah; Frida Soesanti; Miftahuddin Majid Khoeri; Ari Prayitno; Fabiana C. Pimenta; Maria da Gloria Carvalho; Cuno S. P. M. Uiterwaal; Nina Dwi Putri

Drs. D Safari et al collected clinical isolates of GBS from 177 pregnant women in Jakarta, Indonesia and performed serotying and drug susceptibility tests. This is a valuable manuscript because information from Southeast Asia is rare and well-written. There is no ethical and methodological problem and the information in this manuscript is very valuable.

I suggest one point. This manuscript lack description concerning the situation of prevention method in Indonesia. In Indonesia, is the intrapartum prophylaxis performed? Please mention the situation of GBS infection prevention method in Indonesia in the revised manuscript.

6. PLOS authors have the option to publish the peer review history of their article (what does this mean?). If published, this will include your full peer review and any attached files.

Reviewer #1: **Yes: **Dr Guduru Gopal Rao

Reviewer #2: **Yes: **Timothy Barkham

Reviewer #3: No

---

## [Author Response · Author response to Decision Letter 0]

4 May 2021

REBUTTAL LETTER

Editor:

1. Methodology used needs more clarifications and discussions needs improvements.

RESPONSE: Thank you for your comments. we have revised the methodology and discussion sections.

2. RESPONSE: We changed our financial disclosure and we included our statement in our cover letter.

RESPONSE: We revised our figure in revised manuscript

RESPONSE: We agreed to deposit our laboratory protocols in protocols.io.

RESPONSE: We have reviewed and revised the manuscript and meets PLOS ONE's style requirements

6. We note that the grant information you provided in the ‘Funding Information’ and ‘Financial Disclosure’ sections do not match.

RESPONSE: We changed our financial disclosure and we included our statement in our cover letter.

Financial Disclosure:

This study was partly funded by Partnerships for Enhanced Engagement in Research (PEER) program, the U.S. Agency for International Development with project title “Effect of Air Pollution in Early Life on Infant and Maternal Health Project” (https://sites.nationalacademies.org/PGA/PEER/PEERhealth/PGA_161463) and Ministry of Research and Technology, Republic of Indonesia/National Research and Innovation Agency. This work was supported in part by the U. S. Global Health Security funds and Publikasi Terindeks Internasional (PUTI) Universitas Indonesia. The funder had no role in study design, data analysis, decision to publish and preparation of the manuscript.

CDC disclaimer

The findings and conclusions in this report are those of the authors and do not necessarily represent the official position of the Centers for Disease Control and Prevention.

7. Additional Editor Comments:

The reviewers have raised a number of questions on methodology used and made suggestions for improvements in interpretation and discussion. Please address all reviewer comments point by point.

RESPONSE: We have carefully studied the comments, questions, and remarks which were made by the editor and reviewers. Below is a detailed, point-by-point reply indicating how, why and where we have revised the manuscript

Reviewer #1: 

This limited study is one of many studies describing GBS prevalence in pregnant population but one of the few from Indonesia. It is original in so far that describe the prevalence, serotypes and antimicrobial susceptibility in this population. The study is a part of a wider study 'Effect of Air Pollution in Early Life on Infant and Maternal Health Project'. It appears to be a single centre study. Statistics: No sample size calculation for estimating GBS prevalence undertaken. Thus it is not clear how the sample size was chosen.

RESPONSE: Thank you for your comments. We have revised the manuscript to include analysis of power and sample size (lines 172-175). 

……Power of the test was computed using Stata Software. Previous study found the prevalent of GBS colonization in pregnant woman in Indonesia was 31% (P0 = 0.31) [10] and we estimated an increase of 10% (=0.1) therefor Pa was 41%. Using these parameters, we found the power of the test for a sample size of 179 with alpha 0.05 was 74.3%......

Microbiology: No rectal swabs taken. So likely to underestimate rates.

RESPONSE: We agreed with reviewer – we included this information in limitation of study section (Lines 297-298).

Swabs taken and stored at -80C. However, no assessment of viability of GBS. Authors have not stated how long swabs were stored before culture.

RESPONSE: We used 5 months vaginal swab specimen in STGG media stored in -80oC with no initial testing to assess viability of GBS. However, during the isolation, we boosted the growth of GBS by adding rabbit serum as supplement during enrichment with the selective media LIM broth. We expected that the addition of rabbit serum followed by chromogenic agar plate culturing would boost GBS growth and help isolation. we have added this information as limitation of this study in revised manuscript (Lines 299-303).

 It is not clear how resistance to clindamycin was determined. It is unusual for clindamycin resistance rates to be higher than erythromycin resistance rates. Was inducible resistance tested? 

RESPONSE: Thank you for your response. We performed the antimicrobial susceptibility testing with microdilution using customized 96well plate provided by Thermo with catalogue number GAS-GBS CMC5CDCS. The clindamycin resistant was determined by interpreting MIC value obtained to MIC breakpoints in CLSI. The clindamycin inducible-resistant was provided in MIC plate used in this study. We have added this information in revised manuscript (Lines 157-163 and Lines 223-227).

The discussion is satisfactory but suggest rephrase the sentence in lines 152/3 to state the rate reported in this study may be an underestimation as only vaginal swabs were collected.

RESPONSE: we have added a sentence describing the specimens used in this study are vaginal swab only (Lines 282-283).

There is no section describing the limitations of the study. Finally, the findings of this limited study may not be generalizable.

RESPONSE: We agreed with reviewer regarding the limitations of the study. We revised the manuscript (Lines 297-303).

Reviewer #2: 

Overall – a straightforward study, with findings similar to previous data from Indonesia in 2010. While serotypes are interesting as potential vaccine targets, MLST or WGS would give more discriminating data, allowing more discriminating molecular epidemiology. The data need a statistical analysis.

RESPONSE: Thank you for your suggestion regarding the further GBS study in Indonesia especially MLST and WGS. We agreed with reviewer for data analysis – Please see Table 1. and Figure 1.

> Title – ok

> Abstract – ok

> Main text

1. Line 43 – I don’t consider GBS a ‘diplococcus’; surely just a coccus, or cocci in chains.

RESPONSE: Thank you for your comments. We revised it (Line 49)

2. Line 49/50. I don’t understand this sentence. Perhaps you mean ‘... especially in Africa, where 54% of estimated cases and 65% of all 50 fetal/infant deaths occur.’

RESPONSE: Thank you for your suggestion. We revised it. (Lines 55-56)

3. Line 56; this is a very long sentence, and therefore hard to read. I suggest you break it up into 2 sentences. ‘In general, the prevalence of maternal GBS colonization is 18% worldwide, with considerable regional variation (11-35%). Almost all (98%) colonizing GBS are serotype I-IV, whereas serotype III is associated with invasive disease [5].

RESPONSE: We have revised the statement as below (Lines 62-64)

……..” In general, the prevalence of maternal GBS colonization was 18% worldwide with regional variation (11-35%) and 98% of identified colonizing GBS isolates were serotype Ia, Ib, II, III and IV worldwide whereas serotype III was associated with invasive disease”…..

4. Line 57; do you mean serotypes I and IV? .. or serotypes Ia, Ib, II, III and IV?

RESPONSE: We have revised the statement as below

……..” In general, the prevalence of maternal GBS colonization was 18% worldwide with regional variation (11-35%) and 98% of identified colonizing GBS isolates were serotype Ia, Ib, II, III and IV worldwide whereas serotype III was associated with invasive disease”…..

5. Line 60 -61. This sentence is not clear. Does this data on Ia refer to GBS pairs, isolated from both mother and child pairs? Does it refer to invasive or just colonizing GBS? You must be very specific, otherwise the reader won’t understand your message. Are you telling us that Ia accounts for 40% of invasive neonatal disease? Simply pasting data from a reference without telling us why you are doing so, or interpreting the data for readers, is unhelpful.

RESPONSE: We have revised the sentence. The 40% prevalence is reported for GBS colonization in mother while 38% was vertical transmission prevalence happened between mother and newborns (Line 66-69)

6. Line 75 – change to ... ‘... findings may contribute to new preventive strategies targeting ...’

RESPONSE: We have revised it (Line 83-84)

7. Line 90 – you could delete ‘The content and objectives of the study were explained to the participants and their consent obtained by signature on the appropriate consent forms.’.

RESPONSE: Thank you for your suggestion. We deleted it.

8. Line 92; change ‘eligible’ to ‘consented’

RESPONSE: We changed it

9. Table 1 is confusing, as you have calculated % data in columns. Most readers are interested in the % of positive or negative colonization in each group, meaning the % data should be across rows, not columns. I don’t think the ‘Frequency’ column needs % data at all.

RESPONSE: Thank you for your suggestions. We have revised the Table 1. 

10. Line 126; the text says that there were no differences in rates but I don’t see any statistical analysis (comparing carriage v non-carriage in each category).

RESPONSE: Thank you for your comment. We revised the manuscript with statistical analysis (Lines 167-170). Only univariate statistics are presented when assessing association of GBS with maternal demographics, conditions and outcomes. No confidence intervals provided for rates. We revised the manuscript to include analysis with p value, OR and 95%CI (Lines 214-217 and Table 1 in line 479).

Data Analysis

Data analysis was performed using Stata Software. Bivariate analysis was performed using Chi-Square and Fisher’s Exact to evaluate the association between risk factors to colonization of GBS in pregnant woman. Logistic regression was performed to compute Odds Ratio (OR) with 95% Confidence Interval (CI).

11. Fig 1 does not show any statistical analysis.

RESPONSE: We added the statistical analysis (t-test) in Figure 1 and in the text – Lines 216-219

12. Delete Table 2, as all the data is in the text – lines 138 – 146.

RESPONSE: We deleted Table 2. 

13. Line 149 - 152; a long sentence. It would be easier to read if you cut it up.

RESPONSE: Thank you for your suggestion. We revised it (Line 249-252).

“…Our data showed that the GBS colonization rate in pregnant women was 30% in Jakarta, Indonesia. This rate was higher than the other Asian countries mean rate of 12.8% (country variation: 8%-20%) and also higher than the global rate of GBS colonization (18%-26.2%)…”

14. Line 152 - 153; I think you mean ‘The true colonization rate ...’

RESPONSE: We revised it (line 252)

15. Line 163 – The sentence includes serotype V, but doesn’t include IV, but earlier you said (Line 57), that I-IV accounted for 98% of GBS. Please harmonize.

RESPONSE: Thank you for your input, we have revised it (293-295)

16. Line 189; I think this is a little unfair, as a summary of the reference you cite; they actually say ‘While overall resistance is still relatively rare, a recent study highlighted the increased incidence of PR-GBS in Japan, increasing from 2.3% in 2005–2006 to 14.7% in 2012/13.’ A balanced view is along the lines of 'While there are occasional reports of penicillin resistance, resistance is so rare that penicillin can be safely used empirically'.

RESPONSE: Thank you for your suggestion. We have revised it (Lines 289-290)

 Reviewer #3: 

Manuscript #: PONE-D-21-01468

Title: Prevalence, serotype and antibiotic susceptibility of Group B Streptococcus isolated from pregnant women in Jakarta, Indonesia

> Authors: Dodi Safari; Septiani Madonna Gultom; Wisnu Tafroji; Athiya Azzahidah; Frida Soesanti; Miftahuddin Majid Khoeri; Ari Prayitno; Fabiana C. Pimenta; Maria da Gloria Carvalho; Cuno S. P. M. Uiterwaal; Nina Dwi Putri

 Drs. D Safari et al collected clinical isolates of GBS from 177 pregnant women in Jakarta, Indonesia and performed serotying and drug susceptibility tests. This is a valuable manuscript because information from Southeast Asia is rare and well-written. There is no ethical and methodological problem and the information in this manuscript is very valuable.

RESPONSE: We appreciated your comments. 

I suggest one point. This manuscript lack description concerning the situation of prevention method in Indonesia. In Indonesia, is the intrapartum prophylaxis performed? Please mention the situation of GBS infection prevention method in Indonesia in the revised manuscript.

RESPONSE: We have add the description concerning the situation of prevention method in Indonesia (Lines: 290-296)

…….” At this moment, GBS screening was not part of routine prenatal care in Indonesia. Prophylactic antibiotics were commonly given prior caesarean section [36] and premature rupture of membrane. Recently, it was reported that pregnant women (14.5%) received antibiotics during the perinatal period with start dates ranged between 30 days to several hours prior to labour in Yogjakarta and Central Java Provinces, Indonesia with ceftriaxone and cefotaxime are common antibiotic use in that study [33].

---

## [Decision Letter · Decision Letter 1]

14 May 2021

Prevalence, serotype and antibiotic susceptibility of Group B Streptococcus isolated from pregnant women in Jakarta, Indonesia

PONE-D-21-01468R1

Dear Dr. Putri,

We’re pleased to inform you that your manuscript has been judged scientifically suitable for publication and will be formally accepted for publication once it meets all outstanding technical requirements.

Kind regards,

Iddya Karunasagar

Academic Editor

PLOS ONE

Additional Editor Comments (optional):

All reviewer comments have been addressed satisfactorily. One typographical error has been pointed out by one of the reviewers.

Reviewers' comments:

Reviewer's Responses to Questions

**Comments to the Author**

1. If the authors have adequately addressed your comments raised in a previous round of review and you feel that this manuscript is now acceptable for publication, you may indicate that here to bypass the “Comments to the Author” section, enter your conflict of interest statement in the “Confidential to Editor” section, and submit your "Accept" recommendation.

Reviewer #1: All comments have been addressed

Reviewer #2: All comments have been addressed

Reviewer #3: All comments have been addressed

2. Is the manuscript technically sound, and do the data support the conclusions?

Reviewer #1: Yes

Reviewer #2: Yes

Reviewer #3: Yes

3. Has the statistical analysis been performed appropriately and rigorously? 

Reviewer #1: I Don't Know

Reviewer #2: Yes

Reviewer #3: Yes

4. Have the authors made all data underlying the findings in their manuscript fully available?

Reviewer #1: Yes

Reviewer #2: Yes

Reviewer #3: Yes

5. Is the manuscript presented in an intelligible fashion and written in standard English?

Reviewer #1: Yes

Reviewer #2: Yes

Reviewer #3: Yes

6. Review Comments to the Author

Reviewer #1: The authors have attempted to address the comments of the reviewers. Future studies are necessary to confirm these observations

Reviewer #2: I saw one typo - on line 55/56 - ... 'where 54% of estimated cases and 65% of all 50 fetal/infant deaths occur.

Please remove the '50'.

Reviewer #3: Authors responded to my all comments and the revised manuscript became better. Therefore, I recommend the acceptance for the publication of this journal.

7. PLOS authors have the option to publish the peer review history of their article (what does this mean?). If published, this will include your full peer review and any attached files.

Reviewer #1: **Yes: **G Gopal Rao

Reviewer #2: **Yes: **Timothy Barkham

Reviewer #3: No

---

## [Editor Report · Acceptance letter]

20 May 2021

PONE-D-21-01468R1 

Prevalence, serotype and antibiotic susceptibility of Group B *Streptococcus* isolated from pregnant women in Jakarta, Indonesia 

Dear Dr. Putri:

I'm pleased to inform you that your manuscript has been deemed suitable for publication in PLOS ONE. Congratulations! Your manuscript is now with our production department. 

Kind regards, 

on behalf of

Dr. Iddya Karunasagar 

Academic Editor

PLOS ONE